# Measuring Sedentary Behavior by Means of Muscular Activity and Accelerometry

**DOI:** 10.3390/s18114010

**Published:** 2018-11-17

**Authors:** Roman P. Kuster, Mirco Huber, Silas Hirschi, Walter Siegl, Daniel Baumgartner, Maria Hagströmer, Wim Grooten

**Affiliations:** 1Division of Physiotherapy, Department of Neurobiology, Care Science and Society, Karolinska Institutet, 141 83 Stockholm, Sweden; maria.hagstromer@ki.se (M.H.); wim.grooten@ki.se (W.G.); 2Institute of Mechanical Systems, School of Engineering, ZHAW Zurich University of Applied Sciences, 8401 Winterthur, Switzerland; baud@zhaw.ch; 3Institute of Energy Systems and Fluid Engineering, School of Engineering, ZHAW Zurich University of Applied Sciences, 8401 Winterthur, Switzerland; hubermir@students.zhaw.ch (M.H.); silas.hirschi@hotmail.com (S.H.); siew@zhaw.ch (W.S.); 4Function Area Occupational Therapy and Physiotherapy, Allied Health Professionals, Karolinska University Hospital, 141 86 Stockholm, Sweden

**Keywords:** active sitting and standing, artificial intelligence, calibration study, decision tree, electromyography, inactive sitting and standing, indirect calorimetry, objective measurement, occupational physical activity and sedentary behavior monitor, sensitivity and specificity

## Abstract

Sedentary Behavior (SB) is among the most frequent human behaviors and is associated with a plethora of serious chronic lifestyle diseases as well as premature death. Office workers in particular are at an increased risk due to their extensive amounts of occupational SB. However, we still lack an objective method to measure SB consistent with its definition. We have therefore developed a new measurement system based on muscular activity and accelerometry. The primary aim of the present study was to calibrate the new-developed 8-CH-EMG+ for measuring occupational SB against an indirect calorimeter during typical desk-based office work activities. In total, 25 volunteers performed nine office tasks at three typical workplaces. Minute-by-minute posture and activity classification was performed using subsequent decision trees developed with artificial intelligence data processing techniques. The 8-CH-EMG+ successfully identified all sitting episodes (AUC = 1.0). Furthermore, depending on the number of electromyography channels included, the device has a sensitivity of 83–98% and 74–98% to detect SB and active sitting (AUC = 0.85–0.91). The 8-CH-EMG+ advances the field of objective SB measurements by combining accelerometry with muscular activity. Future field studies should consider the use of EMG sensors to record SB in line with its definition.

## 1. Introduction

Sedentary Behavior (SB) is the most frequent human behavior during waking hours [1,2]. Research has linked SB to a plethora of serious chronic lifestyle diseases like diabetes and metabolic syndrome as well as premature death [3]. According to OECD statistics, up to 72% and 81% of the European Union and United States population worked in the tertiary, predominantly office-based sector in 2016 [4]. Office workers, in particular, spend a high amount (around 75%) of their working time sedentary and only around 8% in non-static activities, and are thus at an increased risk to suffer from SB-related health consequences [5,6,7]. Contrary to expectation, the SB-related mortality risk (1.2–1.4) seems to be higher than for tobacco (1.18) and alcohol (1.02) [7,8], and remains significant even after adjusting for physical activity (PA) [3,9,10].

Slogans like “sitting is the new smoking” have been introduced with the intention to reduce occupational SB [11]. However, a recent review by Stamatakis et al. highlights that there is still a huge lack of knowledge regarding SB, and we are far from being able to create evidence-based public health recommendations [12]. Hence, due to the high prevalence and adverse health effects associated with SB, it is of upmost relevance to study SB and the effects of methods for creating active desk-based office work, so called active workplaces. These typically include sit-stand desks and/or dynamic office chairs [13,14]. It is currently recommended to use these active workplaces [15,16] even though we lack evidence whether they have an impact on energy expenditure [13,14]. For these reasons, there is a need to develop new objective methods to measure SB in field settings.

According to the commonly used SB definition (“*any waking behavior characterized by an energy expenditure ≤ 1.5 metabolic equivalents* (METs), *while in a sitting or reclined posture*” [17], p. 540), body posture and MET must be considered at the same time. A device measuring occupational SB must be able to: (1) distinguish sitting/reclining from standing and other activities (Posture Classification), and (2) be able to classify MET while sitting into below or above 1.5 (MET Classification).

However, the typically used SB measurement set-up in epidemiological studies uses a hip placed ActiGraph^®^ accelerometer and a cut-point of 100 counts per minute to separate SB and light PA [18]. This cut-off point is more of a pragmatic approach than evidence based and not consistent with the definition of SB. When placing the sensor at the hip, every sitting condition in which the pelvis remains in direct contact with the seat is classified as SB, regardless of the actual MET. A similar miss-classification has been described for cycling [19]. Furthermore, the counts per minute measure does not consider body posture [20]. In contrast, another frequently used SB measurement set-up uses a thigh-worn ActivPal^®^ to distinguish sitting from standing [20]. This set-up has proven its accuracy in measuring posture, but it also lacks an accurate MET classification [21,22].

During SB, we assume that most of the energy expenditure on top of the resting metabolic rate (RMR) is spent on keeping body posture and moving body segments. Accordingly, the measurement of muscular activity as primary energy-consuming tissue would be the most direct way to classify MET level consistent with the definition of SB. However, existing devices measuring muscular activity (electromyography systems, EMG) are not suitable for long-term field recordings. This is why we built a new, easy to use 8-channel EMG plus accelerometer system (8-CH-EMG+): The accelerometer is used for posture classification, and the EMG additionally for MET classification. The present study aimed to calibrate the 8-CH-EMG+ for posture and MET classification against an indirect calorimeter during typical desk-based office work in healthy participants.

## 2. Materials and Methods

### 2.1. Participants

A total of 25 participants were recruited for this study by mail, flyer, and word of mouth. Healthy volunteers between 18 and 65 years and able to do typical desk-based office tasks were included. We excluded persons: (1) with mental restrictions limiting their ability to consent to the research, (2) with a silicone allergy, (3) with insufficient study language skills to understand the written and oral study information, and (4) with an acute disease.

The mean age (± standard deviation) of the 13 males and 12 females was 31 ± 8 years (between 20 and 61), mean height 175 ± 9 cm (between 155 and 190), mean weight 70 ± 10 kg (between 57 and 93), and mean BMI 22.9 ± 2.5 kg/m^2^ (between 18.5 and 28.7). The study was conducted in accordance with ICH guidelines and the Declaration of Helsinki. Every participant signed an informed consent form prior to study inclusion. The study was conducted in a bi-national project. Ethical approval was obtained in Sweden (Regionala etikprövningsnämnden i Stockholm DNR 2018/554-31/1), while in Switzerland the study does not fall under the Human Research Act.

### 2.2. Tasks and Workplaces

The investigated office tasks are listed in Table 1. All desk-based tasks were performed in sitting and standing position, resulting in a total of nine office tasks of which four were mandatory and completed by every participant: typing while sitting and standing, video sitting, and walking. Each task lasted 5 min. Additionally, each participant completed a 10-min RMR measurement. Participants lay in a supine position on a padded yoga mat and were not allowed to move.

To reflect different workplace designs in real life, the measurements were taken at three workplaces: One was equipped with two 22″ computer screens (*n* = 11), and a second with a 15″ Laptop (*n* = 12). Both were equipped with a height adjustable table (controlled by the participant). The two remaining participants were recorded at a third workplace design (laptop with external screen, separate standing and sitting desk). The results of these two participants were used to get a first idea whether the models learned the study workplaces (lower performance on these participants). Note that data from these two subjects were evaluated exactly the same way as all the others.

### 2.3. Measurement Equipment

#### 2.3.1. 8-CH-EMG+

The 8-CH-EMG+ (Figure 1) consists of 8 pairs of pre-gelled surface electrodes (Ambu^®^ BlueSensor N, Ambu A/S, Ballerup, Denmark) and a 3d accelerometer (ADXL345, Analog Devices Inc., Norwood, MA, USA). The electrodes were placed bilaterally on the most prominent bellies of the muscles of interest as specified in Table 2. Electrodes were placed without skin preparation. The accelerometer was placed after verbal instruction by each participant themselves on the lateral side of the left thigh (the research staff did not correct wrong placement).

Electrodes and accelerometer were connected by cables to two data processing units and a data logger (Raspberry PI, Figure 1). Analogue EMG data was pre-processed as follows: differential amplifier, high pass filter (cut-off frequency: 1.8 Hz), rectifier, low pass filter (cut-off frequency: 110 Hz), A/D-converter, and saved with 30 Hz to a comma separated file (.csv).

#### 2.3.2. Indirect Calorimeter

The indirect calorimeter (cosmed^®^ K5, Rome, Italy) used as the criterion measure was calibrated in the morning of each recording day as recommended by the manufacturer (flowmeter, scrubber, and room air calibration) and kept powered on until the last recording on each day. Data were recorded with a frequency of 0.1 Hz using the mixing chamber mode. The recording unit was remote controlled and data were real-time streamed via Bluetooth to a computer. Energy expenditure data in kilocalories calculated by the manufacturer software (Omnia, Version 1.6.2, cosmed^®^ K5, Rome, Italy) were exported to a comma separated file (.csv).

### 2.4. Procedure

Participants were first briefed about the measurement and signed the informed consent form. Only participants who confirmed that they refrained from eating and drinking sugary, caffeinated, and alcoholic beverages (for 2 h) and refrained from sport (for 12 h) were included in the study. Thereafter, they were equipped with the 8-CH-EMG+ and the indirect calorimeter. While getting familiar with the mask of the indirect calorimeter, participants filled out a questionnaire regarding their personal characteristics, and looked for a video on YouTube^®^ that interests them but does not make them laugh (for the video task). Subsequently, the measurements of the office tasks started (see Section 2.2. Tasks and Workplaces). Each participant completed as many as possible of the tasks before the 45 min recording session expired. Task order was randomized using the randomization function of MATLAB^®^ 2018a (Mathworks Inc., Nattick, MA, USA). However, all mandatory tasks were completed within the first 6 tasks, and most participants completed 7 tasks. The execution of the tasks was not standardized. Participants completed all tasks in their normal working speed and in their own way. For example, they were allowed to talk, to open doors while walking, or to change table and chair height at any time. Directly after completing the office tasks, the EMG system was removed and the RMR measured.

### 2.5. Data Preparation and Evaluation

Comma separated files of both measurement systems were loaded into MATLAB. To express the energy expenditure in MET, each task was referenced to the RMR, defined as the median energy expenditure during the second five minutes lying [27,28]. Each office task was assigned to one MET value (median of the final two minutes).

The 8-CH-EMG+ data was analyzed on the typical minute-by-minute basis as follows: (1) Posture Classification: Classify each minute as either sitting, standing or walking, (2) MET Classification: Classify sitting and standing as either inactive (SB, inactive standing) or active (active sitting, active standing) using the 1.5 MET threshold.

The development of the classifications was done with a supervised machine-learning technique based on artificial intelligence. To limit complexity, model type was limited to a simple decision tree using MATLAB’s fitctree function from the statistics and machine learning toolbox (with Gini’s diversity index as split criterion). The decision tree was chosen because it allows the most relevant features to be identified. Feature selection was split in two parts: First, a pre-selection was made using histograms (inspecting the non-overlapping areas) and correlation coefficients (only for MET classification). Second, an automated stepwise feature inclusion algorithm identified the best performing models with as few features as possible [29]:(1)Build a separate model for each feature(2)Train each model and calculate its validity(3)Select the model with highest validity(4)Add each remaining feature to the selected model, and proceed with (2) until no feature remains.

To calculate the model validity (step 2), a hold-one-participant-out cross-validation procedure was used: train a model with all but one participant (the holdout participant), and calculate the model prediction performance on the holdout participant. Repeat the training until each participant is holdout once, and calculate the area under the ROC curve (AUC) over all holdout participants. This means that each model used n times n−1 participants for training and 1 participant for validation, so that each participant is used to validate each model once (*n* = 25). This procedure ensures that participants included in the training of the model are not also included in its validation and reduces the risk of overfitting. From the selected models (each time the algorithm passes step 3), the one with largest AUC was finally taken, considering a penalty of 10^−3^ for each additional input feature to limit model complexity.

#### 2.5.1. Posture Classification

The model used the known posture of each task as target (sitting, standing, and walking) and the accelerometer signal features as input. The following 10 time-based features for the sensor *y*-axis (along thigh longitudinal axis, pointing upwards while standing) were pre-selected and entered into the stepwise feature inclusion algorithm: mean, median, 5th and 95th percentile, variance, interquartile-range, kurtosis, skewness, mean angle to the vertical axis and its variance.

#### 2.5.2. MET Classification

MET classification was done for sitting and standing separately. The models were set up with the MET category of each task as target (≤1.5 MET, >1.5 MET) and the 8-CH-EMG+ signal features as input. Feature pre-selection resulted in a total of six time-based and six frequency-based features: Standard deviation, one second lag autocorrelation, number of prominent peaks, sum of prominent peaks in relation to the median, number of median crossings and median time between adjacent median crossings, mean frequency, 1st harmonic frequency and its power, average signal power, power above 0.5 Hz and power above 5 Hz. Each feature was calculated for all eight EMG and three accelerometer channels. For the accelerometer, mean, median, 5th and 95th percentile were additionally included. These features were not used for the EMG because EMG raw data is well known to have a large inter-personal variability. In summary, a total of 144 features were pre-selected (12 per EMG channel, 16 per accelerometer axis).

The subsequent stepwise feature inclusion algorithm ran twice: once with AUC, and once with Matthews correlation coefficient (MCC) to assess validity on unseen data. The models with highest AUC and highest MCC are presented, including global (AUC, MCC, Sensitivity, Specificity, and Positive Predictive Values) and individual summary statistics (AUC and MCC with 95% confidence intervals). Additionally, the model with highest AUC and MCC which requires no more than 3 EMG channels is also presented. In case the combination of the presented models increases the classification performance, they are also presented.

All reported average measures like ambient temperature and RMR were checked for violation against the normal distribution using Lilliefors test (significance level set to 0.05). Mean ± standard deviation and median (interquartile range) were used for negative and positive test outcomes, respectively.

## 3. Results

In total, the 25 participants performed 168 tasks, of which were 379 min performed in sitting, 331 min in standing, and 121 min in walking. A total of 9 min (1.1%) were lost due to unintentional EMG recording stops. The mean ambient temperature was 26.8 ± 0.9 °C (between 25.1 °C and 28.3 °C), and mean relative humidity was 49.6% ± 5.3% (between 38.5% and 58.6%). Median RMR was 2011 (802) kcal/day. Average MET per task and number of minutes spent inactive are presented in Table 3.

### 3.1. Posture Classification

The best performing decision tree used the 5th and the 95th percentile of the accelerometers *y*-axis and classified all minutes correctly into sitting, standing and walking (AUC = 1). The final tree including the measured values is shown in Figure 2.

### 3.2. MET Classification

#### 3.2.1. SB and Active Sitting

Participants spent 87% of the sitting time sedentary (MET ≤ 1.5) and 13% active. The best performing models classified 85–93% of all minutes correctly (Table 4). Model AUC-11 had the highest AUC, model MCC-10 the highest MCC, and model AUC-6 the highest AUC and MCC with only 3 EMG channels. Performance increased to 90–95% if the models were combined (AUC-6 & MCC-10, AUC-11 & MCC-10). A complete list of all included signal features as well as the developed decision trees can be found online in the Appendix A.

#### 3.2.2. Inactive and Active Standing

Participants spent 86% of the standing time inactive (MET ≤1.5) and 14% active. The best performing models classified 90–93% of all minutes correctly (Table 4). One single model was identified with highest AUC and MCC (AUC-8), and one single model with highest AUC and MCC with only 3 EMG channels (AUC-5).

## 4. Discussion

This study calibrated the 8-CH-EMG+, a new measurement device combining accelerometry with muscular activity, to measure posture and activity during typical desk-based office work. By means of a machine learning data processing technique, we developed a set of simple decision trees, which first classify posture and subsequently the activity level of sitting and standing. The models classified between 85% and 95% of all minutes correctly into (in)-active sitting and standing. For SB, the best performing model (highest AUC) had a sensitivity of 89.1% and specificity of 93.9% (AUC-6 & MCC-10). Unfortunately, these numbers cannot be compared to those from other studies in a straightforward way. Most calibration and validation studies investigating SB use sitting as a proxy for SB [30,31,32,33,34,35,36] and neglect active sitting (13% of all sitting minutes in this study, Table 3). The same happens if one looks at the compendium of physical activity [37] to classify a particular office task on the group level into SB or not, for example [38]. It typically means that all sitting tasks are classified as SB, and thus the SB classification problem turns out to be a posture classification problem. The sensitivity and specificity presented in such studies (for example 92.8% and 99.3% for the SIP method in [38]) must therefore be compared to the performance of the posture classification algorithm of the present study (sensitivity and specificity of 100%). Unlike previous studies, the primary aim of this study was to classify each task for each individual separately into SB consistent with its definition (≤1.5 MET while in a seated posture). We therefore first ran the posture classification algorithm, and minutes correctly classified were subsequently used for the activity classification. This architecture allows completing the model with other activity classifications like moderate and vigorous PA, while the SB classification remains valid.

### 4.1. Posture Classification

The high performance of thigh-worn accelerometers for posture classification is well known, which is why we limited this analysis to accelerometer data [20,21,22,34]. The presented decision tree solved the posture classification problem using thigh orientation and acceleration/deceleration perfectly: First, the tree separates sitting from upright activities using the 5th percentile of the thigh-mounted accelerometers longitudinal axis. The presented number (≥−0.774) is equal to a 39° thigh angle to the vertical axis (if assuming a quasi-static posture). If the 5th percentile is larger (less negative), the minute is assigned to sitting. Thus, the tree requires a more horizontal thigh orientation than 39° during 95% of the time to classify a minute as sitting.

If this is not the case, the tree separates walking from standing using the dynamic thigh motion of each step: If the acceleration (5th percentile more negative) or deceleration (95th percentile less negative) exceed a certain threshold, the minute is assigned to walking, else to standing. Thus, to be classified as walking, a minute must contain a certain amount of thigh acceleration (second tree level); or a certain amount of deceleration (third tree level), but we do not know the precise amount (for example how many steps). However, all minutes with only a few steps (as observed during the deskwork task in standing) were correctly classified as standing, while all walking minutes, even those with short breaks to open doors, were correctly classified as walking.

### 4.2. MET Classification

To separate SB and active sitting, we present five decision trees (Table 4), each of them has specific pros and cons. Roughly spoken, the models can be separated in two categories: those more suitable to detect SB (sensitivity ≥ 93%), and those more suitable to detect active sitting (specificity ≥ 94%). In both categories, the model with higher AUC and MCC rely on more EMG channels. Thus, if the number of EMG channels is of relevance (for example in a large scale epidemiological study), model AUC-6 might be the best choice. This model depends only on three EMG channels (right forearm, left upper arm and shoulder) but still has a sensitivity of 93%. However, if the number of EMG channels is less important than the model performance, we recommend using the combined AUC-11 & MCC-10 model with six EMG channels and 98% sensitivity. In case both, SB and active sitting are of equal interest, model AUC-6 & MCC-10 might be the best choice (89% sensitivity and 94% specificity).

In addition to the activity classification for sitting, we also present an activity classification for standing. Previous studies report inconsistent findings whether standing causes a higher MET than sitting [24,39,40,41], and our data set displays no fundamental MET differences between sitting and standing. It therefore remains unclear whether a workplace intervention that reduces sitting also affects the activity level. Future field studies might therefore want to examine the activity level of standing. The automated feature inspection algorithm resulted in only two models: one with three EMG sensors (AUC-5), and one with four EMG sensors (AUC-8). Both models have a high sensitivity (≥94%) and positive predictive value to detect inactive standing (≥94%), but a limited performance on active standing (66% and 70%). Again, model choice should be influenced by the number of EMG channels it is feasible to record.

### 4.3. Critical Appraisal

All recordings of this study were made out-of-the-lab at three typical office workplaces to account for real-life data. The majority of data was recorded on two different workplace designs, and the third was used to get a first idea whether the models learned the study workplaces, which seems not to be the case as we found no fundamental different classification performance for the two subjects recorded at the third workplace design. However, this study must be seen as a first step towards the use of EMG sensors to measure SB, and a detailed analysis of workplace design should be made in a subsequent study, including also different participants and office tasks. Task selection was informed by previous research in this field and the execution was not standardized. Participants performed each task in their own way, and we noticed fundamental differences between participants (for example for deskwork: some left the folder on the side of the table and moved forth and back between the folder and the computer, others took the folder to the middle of the table). In addition, the door of the office was kept open for all office tasks but not the RMR measurement. Due to the noise from the corridor, there were some additional environmental factors for the participants that we have not controlled, just as in real offices. We do not know how this affected our results, but we strongly believe it increases the ecological validity of the presented models.

This study measured RMR instead of using the standard oxygen consumption (3.5 mL/min/kg). Accordingly, the presented MET values reflect the actual relation between energy expenditure at rest and at work, and there is no need to measure a participant’s body weight in future studies to calculate RMR. Using the self-reported participant weight, our sample has on average an oxygen consumption of 4.1 ± 1.0 mL/min/kg at rest. Thus, if we had taken the standard oxygen consumption, our MET value would have been 17% higher and significantly more minutes would have been classified as active. A potential explanation for the increased relative oxygen consumption in our sample could be an underestimation in the self-reported body weight, or the high mean room temperature (26.8 °C). However, all tasks were recorded at the same temperature, which might eliminate temperature effects, and body weight was not used to develop the models.

The data processing of this study was done without any data cleaning. Even if one sensor partially lost its connection, the sensor was not removed from the analysis. In field use, partially missing data is one of the biggest challenges for a multi-sensor-system, and thus including it in the calibration is the best way to account for this issue. However, it is likely to have reduced the model performance because features of partially missing channels (consecutive zeros) have a limited classification ability, and are therefore not selected by the automated feature inclusion algorithm. Two important aspects of the typical EMG usage in scientific studies were not considered in this study with respect to a later field use: skin preparation and reference contraction. We assume that participants of a future field study who might receive the device for example by mail would probably not properly prepare their skin (for example cleaning, shaving) and are not familiar with maximum voluntary contractions, which is why we omitted both. However, one participant had very hairy forearms where the electrodes did not stick on the skin at all, and was therefore shaved. The non-standardized raw EMG signals are the reason why we have not used absolute EMG features like mean or percentiles. Since we were not interested in the activity of a single muscle, we did not consider crosstalk.

We consider the model development in this study as one of its biggest strength. As opposed to other studies using artificial intelligence data processing techniques, an objective method to include signal features was used, and the number of features as well as model complexity was restricted. The typical feature selection is informed by previous studies, authors’ experience and extensive feature lists, resulting in a large number of features (for example 148 for a single 3d accelerometer [42]). Starting also from such a large feature list, we first pre-selected relevant features by inspecting histograms and calculating correlation coefficients. The pre-selection, also known as feature filtering, simplified further data processing at the potential cost of biasing the model performance upwards. However, adding a random selection of features excluded in the pre-selection in the stepwise feature inclusion algorithm does not improve model performance. From the original 309 features, 144 were included in the automated feature inclusion algorithm. The algorithm treated each feature for each EMG and accelerometer channel separately. This procedure was key to end up with models having no more than 11 features. We compared our final models with models using predefined feature selection, and the automatically derived models always outperformed the predefined. We therefore strongly recommend future calibration studies using an automated feature inclusion algorithm, and to treat the features of each channel separately. However, such an algorithm is computationally demanding and has its disadvantages: It takes in each round the feature that performs best, ignoring feature combinations that might perform even better. To better account for such interaction effects, one could use a feature exclusion algorithm (starting with all features, and iteratively remove one). However, such an algorithm is computationally much more demanding as one starts with n models having each n − 1 features (if n is the number of input features, *n* = 144 in this study). For our automated feature inclusion algorithm, it is worth mentioning that although the algorithm included in each round one additional feature, the outputted decision tree did not necessarily use all of the inputted features. For example, model AUC-11 required 14 passes through the algorithm and thus had 14 signal features as input, but the final decision tree depended only on 11 (three features were not used). We assume that this is the result of feature interaction effects.

We limited model complexity by inspecting only simple binary decision trees. Our final models do therefore not depend on artificial intelligence data processing techniques, and can be used even by researchers without artificial intelligence computer skills. However, we used artificial intelligence to develop the models, and it remains unknown whether we would have observed similar results with traditional statistical approaches. The presented model performance might be higher if we had taken more advanced models like random forest classifiers or support vector machines. However, the slightly improved overall performance would come along with higher complexity, and future studies depending on our processing software (MATLAB) to use the model. Instead of taking a sophisticated model, we decided to keep the model simple and improve the feature inclusion algorithm to identify the most relevant and powerful ones. A complete list of all relevant signal features as well as the developed decision trees can be found online in the Appendix A. If we understand the relationship between muscular activity and MET while desk-based office work in more detail, it might be worth to re-analyze our data with more sophisticated machine learning modelling techniques.

Although the 8-CH-EMG+ prototype is independent and allows recording for up to 8 h, we recommend updating the device by incorporating the electronics into a smart textile for future field use. Such a smart textile with built-in dry electrodes has already been successfully used to measure the EMG amplitude [43]. A smart textile might make the number of recorded EMG channels irrelevant. However, it will need a detailed data inspection to check signal quality. In this regard, we are currently updating the system to use dry instead of wet electrodes, which is probably the most difficult step towards a portable, user-friendly measurement system for everyday use.

All in all, we believe that the applied data recording out-of-the-lab, the automated feature inclusion algorithm and the simple model make the results of this study promising. However, a future field study using different people, workplaces and tasks will have to investigate whether the typical accuracy loss from calibration to field use (see [35]) also applies to the presented models.

## 5. Conclusions

The result of the present study demonstrates that combining accelerometer data with EMG can provide a deeper insight into the objective measurement of occupational SB and active sitting, as well as into inactive and active standing. All presented models classified between 85% and 95% of the recorded time correctly, even though some included as few as three EMG sensors in combination with an accelerometer. We therefore recommend future field studies to incorporate EMG sensors to monitor SB objectively consistent with its definition.

## Figures and Tables

**Figure 1 sensors-18-04010-f001:**
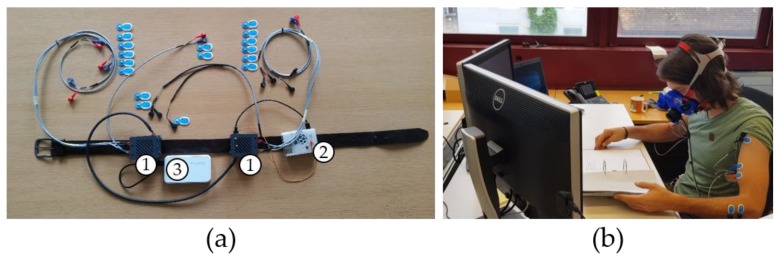
(**a**) 8-CH-EMG+ prototype with electrodes, wires, two data processing units (1), data logger (2), and power bank (3). (**b**) Participant performing the deskwork task with 8-CH-EMG+ and indirect calorimeter (sleeve rolled up to show the electrodes).

**Figure 2 sensors-18-04010-f002:**
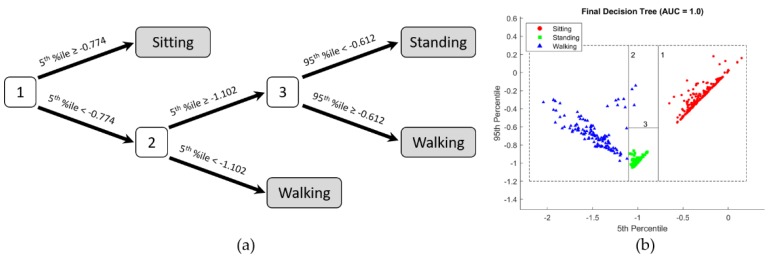
(**a**) Decision tree to classify each minute into Sitting, Standing, and Walking using the 5th and 95th Percentile (%ile) of the accelerometer *y*-axis (accelerometer attached to the left thigh, *y*-axis pointing upwards while standing). 99.6% of all minutes were correctly classified using the first two nodes of the decision tree, and 100% using all three nodes. (**b**) The actual measurements, including lines and numbers to visualize the decision tree nodes.

**Table 1 sensors-18-04010-t001:** Investigated office task. Task selection was based on previous studies investigating typical office work [13,23,24,25,26].

Task	Instruction and Aim
Video ^1^	Watching a self-determined video on YouTube^®^ to investigate listening and watching without own intervention (passive sitting, for example as on meetings).
Mouse ^1^	Playing a computer game with the mouse (Moorhuhn DeluXe, Version 2.6.1.28, Doyodo Entertainment GmbH) to investigate intensive mouse use.
Typing ^1^	Writing a text on the computer to investigate intensive keyboard use.
Deskwork ^1^	Doing various predefined short tasks with a folder and an excel file (get a folder and search in it, do mental arithmetic, create tables, write notes, switch screen views) to investigate successive short office tasks with and without computer [26].
Walking	Walking around on the floor with a weight of 800 g in one hand (for example a tablet, a book or a bottle) to train the system to detect non-stationary office activities like walking to the printer or in a meeting in order to exclude them from MET classification

^1^ All desk-based tasks were performed in sitting and standing.

**Table 2 sensors-18-04010-t002:** Instruction for the electrode placement of each EMG channel (CH) including primary motion that the sensor detect and the corresponding muscle names.

CH	Electrode Placement	Primary Motion	Muscle
1	Dorsal on the proximal half of the forearm, on the muscle belly that moves the most when the participant is typing with the fingers in the air	finger and wrist	extensor digitorum
2	Frontal on the middle part of the upper arm, on the most prominent muscle belly when the participant flexes the elbow 90°	lower arm	biceps brachii
3	Frontal on the shoulder, on the most prominent muscle belly when the participant flexes the shoulder 90°	upper arm	deltoideus pars anterior
4	Lower back, on the most prominent muscle belly when the participant stands on the contralateral leg	upper and lower body (for example leaning forward, shifting body weight)	erector spinea
Ref	Right iliac crest, on the most prominent bony landmark	N/A	N/A

Note that electrodes were placed bilaterally with approx. 1 cm inter-electrode distance (Figure 1), except the single reference electrode (Ref). N/A: Not Applicable.

**Table 3 sensors-18-04010-t003:** MET value for each task, with number of participants per task (# participants), number of minutes per task (# minutes), and percentage of time spent ≤ 1.5 MET (either sedentary behavior or inactive standing).

	Video		Mouse		Typing		Deskwork		Walking
	Sitting	Standing	Sitting	Standing	Sitting	Standing	Sitting	Standing	
MET ^1^	1.09 (0.22)	1.13 (0.20)	1.23 (0.23)	1.29 (0.22)	1.24 (0.26)	1.24 (0.23)	1.46 (0.24)	1.47 (0.39)	2.87 (0.74)
# participants	22	13	13	14	25	25	16	15	25
# minutes	110	64	65	70	125	125	79	72	121
% ≤ 1.5 MET	100	100	77	100	96	88	63	56	0

^1^ MET: Metabolic Equivalent, indicated is the median (interquartile range).

**Table 4 sensors-18-04010-t004:** Classification performance to separate inactive from active behavior for sitting and standing (inactive ≤1.5 MET). The models are named according to the used optimization criteria and the number of features included. Results appear separately on global (study population) and individual (participant-by-participant) levels. The models were optimized on the global level. The right part of the table shows the number of EMG channels per model. Numbers in bold mark the reason why a particular model was selected (highest AUC or MCC, lowest number of EMG channels). A complete list of all included signal features of each model as well as the model itself can be found online in the Appendix A.

		Model Performance											EMG Channels								
		Global Level							Individual Level				Right Side				Left Side				
		AUC	MMC	Perf	Sensitivity	Specificity	PPV INACT	PPV ACT	AUC [CI_95%_]		MCC [CI_95%_]		1	2	3	4	1	2	3	4	#
Sitting	single models																				
	AUC-6	0.85	0.65	91.3%	93.3%	77.6%	96.6%	63.3%	1.00	[0.967 1]	1	[1 1]	x					x	x		**3**
	AUC-11	**0.90**	0.60	84.7%	82.7%	98.0%	99.6%	45.7%	0.97	[0.900 1]	1	[1 1]	x		x	x		x	x		5
	MCC-10	0.86	**0.70**	93.1%	95.8%	75.5%	96.3%	72.5%	1.00	[0.967 1]	1	[1 1]			x	x		x	x	x	5
	combined models																				
	AUC-6 & MCC-10 *	**0.91**	0.68	89.7%	89.1%	93.9%	99.0%	56.1%	0.97	[0.960 1]	1	[1 1]	x		x	x		x	x	x	6
	AUC-11 & MCC-10 **	0.86	**0.77**	95.0%	98.2%	73.5%	96.1%	85.7%	1.00	[0.990 1]	1	[1 1]	x		x	x		x	x	x	6
Standing	single models																				
	AUC-5	0.80	0.59	89.7%	93.7%	66.0%	94.3%	63.3%	1.00	[0.989 1]	1	[1 1]	x	x			x				**3**
	AUC-8	**0.84**	**0.71**	93.4%	97.2%	70.2%	95.2%	80.5%	1.00	[0.942 1]	1	[1 1]	x	x	x		x				4

* If one of the models predicts active sitting, the minute is assigned to active sitting. ** If one of the models predicts sedentary behavior, the minute is assigned to sedentary behavior. Abbreviations: AUC (Area under the ROC curve), MCC (Matthews correlation coefficient), Perf (Performance, percentage of correctly classified minutes), PPV (IN)ACT (Positive predictive value for (in)active behavior), CI_95%_ (95% Confidence Interval), EMG Channel 1–4 (1: forearm; 2: upper arm; 3: shoulder; 4: lower back, see Table 2), # (Number of channels used by a particular model).

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
