# Peer review of "Measuring Sedentary Behavior by Means of Muscular Activity and Accelerometry"

_sensors, 2018, doi:10.3390/s18114010_

Round 1
Reviewer 1 Report
The authors presented a method to measure human sedentary behavior using EMG and accelerometer. A few suggestions which may help to improve the manuscript.
1. The authors suggested that one of the main contributions of this work is validation of the benefit of including EMG sensors during sedentary behavior analysis. Have they compared the accuracy between: (1) using accelerometer only; (2) using EMG only; (3) using EMG and accelerometer together?
2. Is calibration needed for each individual? Are the trained models more general or more subject-based? Have the authors considered the accuracy of model when apply to other users, other than the 25 subjects in this study?
3. The organization of the chapters could be improved. The presentation could be modified to be easier to read. For example, starting with an overview of the method, followed by the basic physical principle behind the method, and then experiment design, test results, discussion etc.
4. The authors may need to add more discussions regarding the limitations of presented method. Current tasks in the experiment are comparably easy and may not reflect the real situations in normal living condition.
5. What are the percentages of the data out of all recordings, used for training, validation and testing respectively?
Author Response
We thank the reviewer for the critical and valuable input helping us to improve the quality of our manuscript. Please find attached a point-by-point answer to all comments and questions.

Reviewer 2 Report
This is a well-written paper with a sound and easy-to-follow introduction. The paper appears to be a good methodology paper, given the slant of the discussion. The authors ought to make this clear in the introduction by expanding the aim of the study (Line 72). The research problem is re-evaluated in the discussion section in light of the results obtained. The results are promising, and this point is clearly made in the discussion. It would have been good to see how the authors propose to develop their method as a portable user-friendly package for occupational professionals, to complement (or even replace) self-report tools for sedentary behaviour in the workplace.
Comments:
Line 67: What do the authors mean by “We hypothesize that during SB, most of the energy expenditure is spent on keeping body posture…”? Please formulate the hypothesis(ses) clearly.
Line 77 and Line 81: I would like to see more details about the participants. How many males and females took part in the study? The age range is rather broad (20 to 61) but the mean age is 31; what was the age distribution? Did the older participants display a stooped posture?
Line 80: How was “insufficient study language skills” assessed?
Line 89: Typo “in totally”.…”in total”
Line 91: Were there any breaks to interrupt prolonged sitting, not including the 10-minute RMR?
Line 108: “themselves” instead of “their self”
Line 143: Typo “in their own way”
Line 144 and Line 338: Replace acronym “e.g.” with “for example”
Line 147: Typo “were loaded”
Line 249: For ease of reading, could the authors please report the results for “Workplace Design” in a more reader-friendly way.
Line 255: The authors mention “artificial intelligence” for the first time in the discussion. The use of artificial intelligence, which is a crucial aspect in this study, should also be made explicit stated when in describing the machine-learning technique (Line 155).
Line 318: The authors did not give any information as to why the confounding factors were not controlled; it undisputed that confounding variables affect internal validity of measurements. On the other hand, what is the reasoning behind claiming that the confounding factors increased the external validity or do the authors specifically mean ecological validity?
Line 338: It is understood that some subjects received the device by mail – this should be stated in the procedure.
Line 355: Typo “treat”
Author Response

(The authors gave the same response as above.)

Reviewer 3 Report
This manuscript reports research on Measuring Sedentary Behavior by Means of Muscular Activity, Accelerometry and Artificial Intelligence. There are several studies which have reported a similar analysis and hence, authors need to address some of the major concerns before the paper is considered for publication.
Specific comments to each section
Introduction section needs to be further improved with more explanation and literature review.
Authors used EMG data which are collected using a large number of sensors and hence they need to explain how they have addressed cross-talk related to EMG data (because EMG sensors are connected in close proximity).
Authors need to explain how the classification task was carried out. The information about features, training and testing data used for classification is missing.
Authors need to compare their results with other similar studies available in literature.
The classification accuracy needs to be supported with specificity, sensitivity and ROC
Discussion and conclusion section could be further improved.
Author Response

(The authors gave the same response as above.)

Reviewer 4 Report
This manuscript outlines an approach to classifying office workers’ behaviours (using 25 volunteers to perform nine work-related tasks over three workplaces) into sitting (sedentary and active), standing (inactive and active), and walking using decision trees to process data from EMG sensors and accelerometry based on tasks and indirect calorimetry. Performance on the training data was promising (sensitivities 83% and higher, specificities 74% and higher).
While I think that these results are promising, I also feel that they are being overstated here and that a more cautious interpretation is warranted until they are replicated, particularly in different settings. I also feel that the AI aspect is being oversold.
Aside from these points, the modelling methods could be clarified so that a reader would be able to replicate them on their own data. I’ve made some suggestions around this below in my specific comments, along with some suggestions around wording throughout and requests for clarification around certain points and additional information.
While I don’t expect the authors to add these, comparisons with other modelling approaches (including standard statistical approaches) would have been useful to the reader in deciding whether this “package” (device and approach) is worthwhile for their own applications or research.
Line 3: I’d prefer “Decision Trees” rather than “Artificial Intelligence” which will be much more useful to someone searching the literature and avoids what I think is overselling this aspect.
Lines 16–17: Perhaps simply “and is associated with a plethora”
Line 21: “newly-developed”
Lines 23–24: “was subsequently performed using decision trees” (I suggest deleting the AI part as I think this creates the wrong impression here, you are using a simple decision-based classifier which you could have just as easily obtained using LDA or a number of other non-AI approaches).
Line 26: Something is missing here
Line 42: Perhaps simply “and remains significant even after”
Line 46: Perhaps “and we are far from being able to create evidence-based”
Line 47: Suggest “high” (rather than “large”) and “with” (rather than “to” towards the end of the line)
Line 71: “long-term” (rather than “long-time”) or “prolonged”
Line 74: The word “subject” has fallen out of favour so perhaps “participants”.
Line 79: “ability to judge” is rather vague, do you mean “ability to perform the tasks” or “ability to consent to the research”?
Lines 80–82: I would personally consider these results and move them to that section instead (Line 132), although I can also see an argument for leaving them here. Sex or gender should also be included.
Line 81: Technically, range is the difference between the maximum and minimum value (https://en.wikipedia.org/wiki/Range_(statistics)) and I’d suggest “between 20 and 61” or similar instead (and for height and weight).
Lines 81–82: Heights and weights are not terribly useful for a mixture of men and women. BMIs would be more useful.
Line 83: “informed consent form prior”
Lines 95–98: This seems rather odd. Only two such participants would never provide meaningful power to detect any such differences even if they did exist unless they were absolutely huge, which seems implausible. Why were these two participants not included in the full data set given you have two different workplaces in any case? (Which are themselves never compared as far as I can tell).
Line 96: “participants”
Line 97: “participants”
Line 98: “participants”
Line 108: “themselves” (rather than “their self”)
Line 116: “participants”
Table 2: “participants” in several places.
Line 124: “the criterion measure”
Line 132: “informed consent form” and “participants”
Line 134: “participants”
Lines 136–137: I’d move these earlier to before the summary statistics are described (preferably around Line 132).
Line 139: “completed as many as possible of the tasks before the 45”
Line 142: “participants”
Line 143: “in their own way”
Line 144: While this stylistic, I wouldn’t use “e.g.” in a manuscript. I suggest “For example, they were allowed to”
Line 147: “were loaded into”
Lines 156–157: This is circular as powerful classification features will naturally lead to good performance.
Line 158: Somewhere around here the particular library and functions used needs to be explained.
Lines 158–160: If I’m understanding this correctly, this pre-selection process will of course bias the model’s performances upwards and this limitation will need to be described in the discussion.
Line 164: Is Step 2 “Evaluate model” or “Validate Model”? It sounds as if training the model would be included in building it (Step 1).
Lines 168–169: This is fine, but I would consider that a very small penalty for AUC per feature. There are more sophisticated ways of regularizing decision trees, but from above, I appreciate that you are looking for a straightforward option here. Personally, I would have been very tempted to use mixed multinomial logistic regression here and then have the advantages of information criteria (AIC, BIC, etc.) and (if ML and not REML was used) likelihood ratio tests for feature selection or discriminant analysis.
Lines 170–173: This is overly optimistic as cross-validation does not provide an unbiased estimate of expected performance on new data and I’d avoid “subject” as a term again. Perhaps “This procedure ensures that participants included in the development of the model are not also included in its validation and reduces the risk of overfitting. Each participant was used to validate each model once.”?
Line 177: I’d suggest “and were entered into the” avoiding the awkward inputted.
Line 178: Was there a reason for using variance over standard deviation here? Means, medians, percentiles, and IQRs are all in the original measurement units, as are standard deviations but not variances. On the other hand, if you wanted the sample moments, means, variances, skewness, and kurtosis would make sense. A case could, therefore, perhaps have been made to include both standard deviation and variance for these two purposes. Any reason for excluding range?
Line 184: “lag-1second-autocorrelation” is awkward, perhaps “one second lag autocorrelation”.
Line 195: I’d suggest deleting the commas before “which” and after “channels”.
Lines 197–199: There are no reported p-values in the manuscript and so I’m not sure what this is referring to.
Line 201: “participants”
Line 207: “participants” (twice)
Table 3: “participants”
Lines 203–205: Note that if these are means ± SDs, that should be made clear (“average” is ambiguous as it includes means, medians, modes, and other). Note also previous point about “range” being a single number.
Line 224: “participants”
Line 231: “participants”
Line 238: “levels” (add “s”) and “participants”
Line 239: “the global level” (add “the”)
Line 240: “bold” (not “bolt”)
Table 4: Note truncation of “Standing”
Line 243: “models” (add “s”)
Line 244: “models” (add “s”)
Lines 246–247: The coverage of the CI should be stated, presumably 95%.
Lines 250–251: Given there are only two such participants (and again I’d use that word rather than “subjects”), this isn’t terribly reassuring. Their values falling within the 95% (presumably) CI isn’t a valid test of this in any case. Again, why were these two participants not included in the full data set?
Line 254: “during” rather than “while”, or “while performing”.
Line 255: I think that this is an overly grand claim—decision trees are in my opinion in that overlap between machine learning and statistics and nothing has been done here that could not also be done with statistical approaches. I’d suggest deleting this first clause and starting “We developed…”.
Line 259: Perhaps “cannot be compared to those from other studies in a straightforward way”
Line 262: “on the group level” (adding “the”)
Line 268: “ran” (rather than “run”)
Line 294: “scale” (delete “d”)
Line 308: “channels it is feasible to record”
Line 312: This is a much, much too strong claim based on n=2 for the third workplace.
Line 314: “participants” (twice)
Line 315: “left” rather than “let”
Line 318: “participants”
Line 318: Given that “confounding” has a precise statistical definition, perhaps “environmental factors” instead?
Line 318: “offices” (add “s”)
Line 323: “a participant’s” (also adding “a”)
Line 328: Probably just the mean would suffice here, but in any case the meaning of the values (mean and possibly SD) needs to be clear.
Line 332: I’d delete “later” here as this applies to all field work of this type.
Line 338: “participants”
Line 340: “participants”
Line 341: “where” rather than “why”
Line 344: “As opposed to”
Line 345: Sorry, but I still think you’re being overly grand in describing this as an “artificial intelligence” study.
Line 347: “authors’” as this is both plural and possessive.
Line 352: I think you should add a comment “at the cost of biasing the model performances upwards” here or similar elsewhere as the pre-screening, which would have to be on the full data set, will have created such a bias.
Line 355: “treat” rather than “trend”
Line 361: “worth mentioning”
Line 362: “did not necessarily use” (add “did” and delete “d”)
Lines 366–368: To be honest, based on my experience, you would probably also have done just as well with standard statistical approaches.
Line 375: Here or elsewhere you need to make the point that “The results from our analyses, while promising, also need to be replicated using different people, workplaces, and tasks and over longer working durations.” (You allude to these points on Lines 383–384 but I think you could be specific about these aspects.)
Line 376: “allows recording for”
Line 377: “recommend updating the device”
Line 382: I don’t think you can claim that these results are “very robust” just yet given the research to date, but perhaps “make the results of this study promising” would be appropriate.
Author Response

(The authors gave the same response as above.)

Round 2
Reviewer 1 Report
The manuscript is improved in this revised version. Part of the issues have been settled.
Reviewer 3 Report
The authors have addressed all my comments satisfactorily and the paper can be considered for publication.